# Urine C-peptide creatinine ratio can be used to assess insulin resistance and insulin production in people without diabetes: an observational study

Richard A Oram,[1,2] Andrew Rawlingson,[1,2] Beverley M Shields,[1] Coralie Bingham,[2] Rachel E J Besser, Tim J McDonald,[1,2] Bridget A Knight,[1] Andrew T Hattersley[1]

RAO and AR are joint first authors.

[1]NIHR Exeter Clinical Research Facility, University of Exeter Medical School, Royal Devon & Exeter NHS Foundation Trust, Exeter, UK
[2]Royal Devon & Exeter NHS Foundation Trust, Exeter, UK

**Correspondence to**
Dr Richard A Oram;
r.oram@exeter.ac.uk

## ABSTRACT

**Abstract: Objectives:** The current assessment of insulin resistance (IR) in epidemiology studies relies on the blood measurement of C-peptide or insulin. A urine C-peptide creatinine ratio (UCPCR) can be posted from home unaided. It is validated against serum measures of the insulin in people with diabetes. We tested whether UCPCR could be a surrogate measure of IR by examining the correlation of UCPCR with serum insulin, C-peptide and HOMA2 (Homeostasis Model Assessment 2)-IR in participants without diabetes and with chronic kidney disease (CKD).

**Design:** Observational study.

**Setting:** Single-centre clinical research facility.

**Participants:** 37 healthy volunteers and 30 patients with CKD (glomerular filtration rate 15–60) were recruited.

**Primary and secondary endpoints:** Serum insulin, C-peptide and glucose at fasting (0), 30, 60, 90 and 120 min were measured during an oral glucose tolerance test (OGTT). Second-void fasting UCPCR and 120 min post-OGTT UCPCR were collected. HOMA2-IR was calculated using fasting insulin and glucose. The associations between UCPCR and serum measures were assessed using Spearman's correlations.

**Results:** In healthy volunteers, fasting second-void UCPCR strongly correlated with serum insulin ($r_s$=0.69, p<0.0001), C-peptide ($r_s$=0.73, p<0.0001) and HOMA2-IR ($r_s$=−0.69, p<0.0001). 120 min post-OGTT UCPCR correlated strongly with C-peptide and insulin area under the curve. In patients with CKD, UCPCR did not correlate with serum C-peptide, insulin or HOMA2-IR.

**Conclusions:** In participants with normal renal function, UCPCR may be a simple, practical method for the assessment of IR in epidemiology studies.

## BACKGROUND

Insulin resistance has been shown to be a significant predictor for the development of diabetes and for cardiovascular risk.[1] [2]

### Strengths and limitations of this study

- This study uses the clinical research facility setting and samples sent from home to demonstrate that urine C-peptide creatinine ratio (UCPCR) can be used in healthy volunteers.
- UCPCR is compared with other epidemiological measures of insulin resistance such as fasting insulin and Homeostasis Model Assessment.
- UCPCR is not valid in people with chronic kidney disease stages 3–5.

Understanding the epidemiology of insulin resistance is important in the identification of patients at risk of type 2 diabetes (T2D) and vascular disease, and for the study of prevention. The optimum individual method to assess insulin physiology uses glucose disposal rate during hyperinsulinaemic-euglycaemic clamp studies,[3] [4] which require infusions of insulin and glucose and cannot be used at a population level. The minimal model analysis of glucose and insulin levels during intravenous or oral glucose loading allows assessment without the use of intravenous insulin, but still necessitates multiple blood samples.[5] The assessment of fasting insulin alone, or with measures of glucose, have been used as a more simple method to study insulin resistance[6] and have been validated against other more invasive tests.[7] One widely used approach that allows for variation in the fasting glucose is the Homeostasis Model Assessment (HOMA, http://www.dtu.ox.ac.uk/homacalculator/ index.php)[8] [9] that models fasting serum glucose, and insulin or C-peptide levels to calculate the insulin resistance. HOMA requires that a fasting blood sample is taken and the sample is relatively rapidly processed within 24 h.[10] This means an appointment with

healthcare or research staff is still required and this is not always readily available for some large epidemiological studies.

An alternative method to blood sampling, which allows samples to be provided without outside assistance, is to measure urinary C-peptide. C-peptide is secreted in equimolar amounts to insulin, but unlike insulin, it is filtered by the kidney with 5% excreted unchanged in the urine, making urinary measures possible.[11] We have recently demonstrated that C-peptide is measureable, reproducible and stable in urine for up to 72 h in boric acid preservative (allowing postage from primary care or from home).[12] Measuring C-peptide as a ratio against creatinine allows the use of a single-spot urine sample by accounting for dilution in the same way as protein creatinine ratio. In patients with T1D and T2D, 2 h urine C-peptide creatinine ratio (UCPCR) is highly correlated with 90 min serum C-peptide in the standard Mixed Meal Tolerance Test.[13 14] We have also shown that in patients with T2D and mild chronic kidney disease (CKD), the correlation between serum C-peptide and urine is maintained.[15] As fasting serum insulin or C-peptide alone is a helpful marker of insulin resistance in people without diabetes, it may be that UCPCR could also be used in this manner.

If UCPCR can be used in people without diabetes this practical method could allow a large scale, population-based assessment of insulin resistance without needing a blood sample to be taken. We aimed to test whether UCPCR could be used as a surrogate measure of insulin resistance in epidemiological studies by examining the correlation of UCPCR with fasting serum insulin, C-peptide and HOMA2-IR (HOMA2-insulin resistance) in participants without diabetes. As a secondary outcome we tested whether stimulated UCPCR could be used as a marker of insulin secretion during an oral glucose tolerance test (OGTT). We also wanted to see if the correlations were maintained in participants with chronic kidney disease.

## METHODS
### Study participants
*Two groups were recruited from December 2009 to May 2010:*

Thirty-seven healthy controls (22 females) with normal renal function estimated glomerular filtration (eGFR>60 mL min$^{-1}$/m$^{-2}$) and normal glucose tolerance[16 17] were recruited from research volunteer databases in Devon.

Thirty patients (8 females) with normal glucose tolerance and a clinical diagnosis of CKD stage 3 or greater (Modification of Diet in Renal Disease (MDRD) eGFR<60 mL min$^{-1}$/m$^{-2}$) (http://www.renal.org/CKDguide/full/UKCKDfull.pdf) were recruited from general nephrology clinics at the Royal Devon and Exeter Hospital. Patients on renal replacement therapy (either dialysis or transplant) were excluded from the study.

### Clinical sampling
Participants fasted from midnight prior to their visit and emptied their bladder on waking (this first-void urine was not collected). Demographic data, medical and drug history were recorded. Baseline fasting blood samples were collected for routine analysis of glucose, glycated haemoglobin and renal function. A second urine sample (second-void fasting) was collected immediately prior to OGTT for measurement of UCPCR (UCPCR0).[6]

In a standard OGTT (75 g glucose), blood samples were collected at 30, 60, 90 and 120 min. A further urine sample was collected for UCPCR analysis at 120 min (UCPCR120). Blood samples were immediately centrifuged and separated. Serum and urine samples were initially stored at −20°C then transferred and stored at −80°C within 1 week. Serum samples were subsequently analysed for insulin, C-peptide and glucose. Urine samples were analysed for C-peptide and creatinine and UCPCR was calculated.

### Biochemical analysis
Urine and serum C-peptide analysis were performed by electrochemiluminescence immunoassay (Roche Diagnostics E170 C-peptide assay, Mannheim, Germany). All urine samples were prediluted 1:10 with equine serum albumin (diluent multianalyte, Roche Diagnostics). The serum insulin analysis was performed by electrochemiluminescence immunoassay (Roche Diagnostics E170 insulin assay). Glucose and creatinine were analysed on the Roche P800 modular platforms. All analysis was performed in the Department of Chemical Pathology, Royal Devon and Exeter Hospital. eGFR was calculated using four-variable MDRD formula.[18]

### Data analysis
Serial serum C-peptide, insulin and glucose measurements were used to calculate area under the curve (AUC) for each parameter. Insulin resistance (HOMA2-IR) was derived from fasting glucose and insulin (http://www.dtu.ox.ac.uk/homacalculator/index.php). The associations between second-void UCPCR0 and stimulated UCPCR120 with serum C-peptide, insulin and HOMA2-IR were assessed using Spearman's correlations. The analyses were performed separately for the group with CKD and the group without CKD. The data for UCPCR were non-normally distributed so a non-parametric statistical testing was used for analysis.

## RESULTS
A summary of the characteristics of the study group is shown in table 1. Three participants with CKD had serum C-peptide samples that were not analysed due to sampling problems; their results have been included in analyses excluding those involving C-peptide values.

**Table 1** Cohort characteristics.

|  | Normal renal function group | CKD group |
|---|---|---|
| Total participants | 37 | 30 |
| Female | 22 | 8 |
| Age (years) | 50 (29–67) | 65 (52–71) |
| BMI (kg/m$^2$) | 27.0 (23.5–33.0) | 26.4 (24.1–28.6) |
| HbA1c (%) | 5.7 (5.4–6.0) | 5.9 (5.6–6.1) |
| Fasting blood glucose (mmol/L) | 4.8 (4.5–5.1) | 5.0 (4.5–5.3) |
| Creatinine (mmol/L) | 77 (66–84) | 195 (134–231) |
| MDRD eGFR (mL/min/1.73 m$^2$) | 88 (76–101) | 32 (26–46) |

Data are presented as median (IQR).
BMI, body mass index; CKD, chronic kidney disease; eGFR, estimated glomerular filtration; HbA1c, glycated haemoglobin; MDRD, Modification of Diet in Renal Disease.

In participants without renal disease fasting second-void UCPCR0 strongly correlated with serum insulin ($r_s$=0.69, p<0.0001), C-peptide ($r_s$=0.73, p<0.0001) and HOMA2-IR ($r_s$=−0.69, p<0.0001). Age and body mass index (BMI) also correlated with HOMA2-IR ($r_s$=0.50 and 0.52 respectively, p<0.0001 for both). Scatter plots with Spearman's correlations and regression lines are shown in figure 1.

After OGTT, UCPCR120 values were higher than UCPCR0 (3.8 vs 1.0 nmol/mmol, p<0.0001; table 2). UCPCR120 correlated with serum insulin ($r_s$=0.78, p<0.0001) and C-peptide AUC ($r_s$=0.8, p<0.0001). Scatter plots with Spearman's correlations and regression lines are shown in figure 2.

In patients with CKD, median fasting (1.2 vs 0.7 nmol/L p<0.0001) and stimulated (457 vs 294 nmol/L, p<0.0001) serum C-peptide measures were higher than the participants without CKD, but serum insulin levels were not different (7685 vs 6180, p=0.4). Despite the higher level of serum C-peptide, UCPCR0 was not different between the two groups (1.0 vs 0.8, p=0.8) and UCPCR120 was lower in the CKD group (3.8 vs 2.7, p=0.02). This is consistent with a reduced renal clearance of C-peptide.

In contrast to healthy controls, there was no correlation between UCPCR0 and fasting serum C-peptide ($r_s$ 0.17, p=0.4), insulin ($r_s$ −0.17, p=0.4) or HOMA-IR ($r_s$ −0.16, p=0.4) and no correlation between UCPCR120 and C-peptide ($r_s$=−0.09, p=1) or insulin AUC during the OGTT ($r_s$=0.26, p=0.2; figure 3).

## DISCUSSION

Our study suggests that a fasting second-void morning UCPCR could be used as a marker of insulin resistance in participants without diabetes, as long as they are known not to have chronic renal disease. The fact that this test can be performed at home without the assistance of healthcare or research staff offers the opportunity to perform a simple assessment of insulin resistance in large scale epidemiological studies.

UCPCR is not a replacement for established measures of insulin resistance, but is an alternative measure of fasting insulin. Numerous population-based studies have used HOMA to estimate insulin resistance.[9] Similarly, there are many studies using euglycaemic clamps and alternative methods such as minimal model analysis to

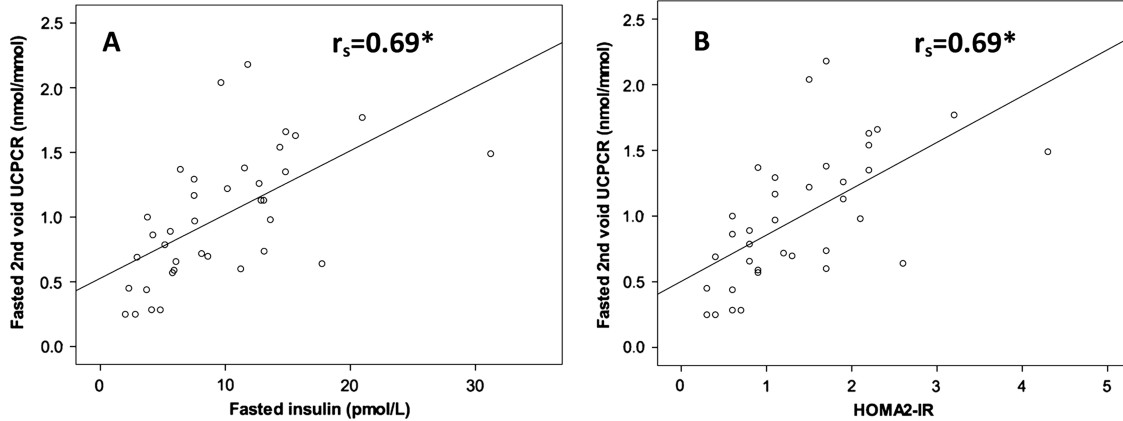

**Figure 1** Scatter plots showing fasting second-void urine C-peptide creatinine ratio (UCPCR0) was strongly correlated with fasting serum insulin (A) and Homeostasis Model Assessment2-insulin resistance (B) in 37 people with normal renal function. Regression line Spearman's $r_s$ correlations are shown. *p<0.0001. Stimulated UCPCR values were correlated with stimulated values of serum insulin and C-peptide, in people without chronic kidney disease.

**Table 2** Median (IQR) serum insulin, C-peptide, UCPCR and HOMA-IR

| | Normal renal function group (n=38) | CKD group (n=30) |
|---|---|---|
| Fasting C-peptide (nmol/L) | 0.7 (0.5–1.0) | 1.2 (0.8–1.6) |
| Fasting insulin (pmol/L) | 8.1 (5.0–13.1) | 8.8 (6.4–12.0) |
| C-peptide area under curve (nmol/L) | 294 (207–405) | 457 (371–550) |
| Insulin area under curve (pmol/L) | 6180 (3641–11 994) | 7685 (5050–9597) |
| UCPCR0 (nmol/mmol) | 1.0 (0.6–1.4) | 0.914 (0.5–1.5) |
| UCPCR120 (nmol/mmol) | 3.8 (2.3–7.0) | 2.8 (0.9–4.0) |
| HOMA2-IR | 1.2 (0.8–1.9) | 1.3 (0.9–1.7) |

Fasting second-void UCPCR strongly correlated with serum insulin, C-peptide and HOMA2-IR in people without chronic kidney disease. CKD, chronic kidney disease; HOMA2-IR, Homeostasis Model Assessment2-insulin resistance; UCPCR, urine C-peptide creatinine ratio.

study individual patients or small groups of patients. UCPCR cannot be used as a direct substitute for these as it only measures C-peptide, and although it shows a strong correlation with HOMA2-IR, we have not validated it against the euglycaemic-hyperglycaemic clamp. The similarity of the scatter plots for UCPCR0 against HOMA2-IR and UCPCR0 against fasting insulin demonstrates the large effect that fasting insulin values have on HOMA-IR when participants do not have abnormal fasting glucose. If the fasting glucose levels are elevated, UCPCR will not correlate so well with HOMA-IR as elevated glucose will start to have an effect on the calculation. This suggests that UCPCR may only be useful as a marker of insulin resistance in populations who have normal glucose tolerance. There was a correlation between HOMA2-IR, age and BMI in our study, but the variance explained by these simple measures was less than UCPCR0 ($r^2=0.27$ for BMI vs $r^2=0.48$ for UCPCR0), suggesting its additional benefit over these measures. UCPCR is a non-invasive test and does not need proximity to a laboratory for immediate sample analysis. Rather than replacing more complex measures of assessment of insulin secretion or resistance, UCPCR is an alternative where serum insulin or C-peptide analysis is impractical, or the non-invasive nature of a urine test is preferred.

We collected second-void fasting urine samples because we have shown this to be less variable than first-void urine in people without diabetes.[12] This is because C-peptide secretion in response to the previous evening's meal will accumulate in an overnight urine sample. A second-void sample adds an extra methodological step which may make sampling more difficult in large studies. It would be interesting to see if first-void urine correlated well with serum insulin and C-peptide and there may be existing studies that have serum and fasting first-void urine samples available to easily test this.

A key finding of this study is that UCPCR0 and UCPCR120 were not correlated with serum C-peptide or insulin in participants with CKD. When comparing the CKD group with the control participants, serum C-peptide AUC was elevated in participants with CKD whereas UCPCR120 was lower. This is explained by the reduced renal clearance of C-peptide in CKD,[11] leading to higher C-peptide AUC values and lower UCPCR120 values. This impaired clearance may then explain the lack of correlation in participants with CKD. The number of patients in this study was too small to compare the patients with different levels of GFR, underlying causes of CKD and the presence of

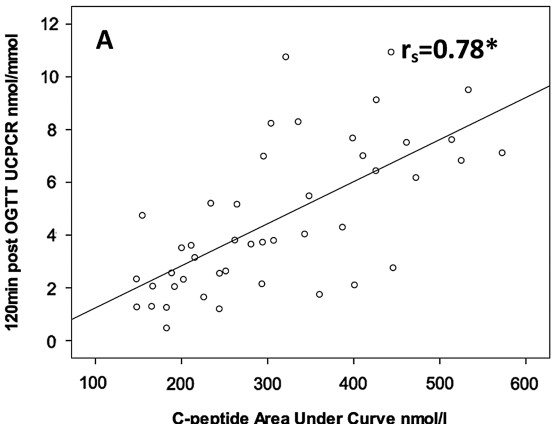
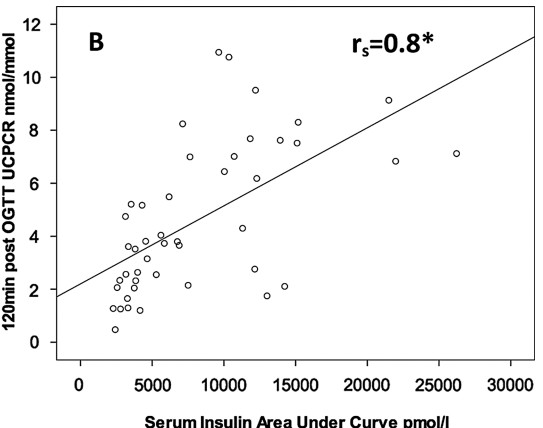

**Figure 2** Scatter plots showing 120 min post oral glucose tolerance test, urine C-peptide creatinine ratio (UCPCR120) was strongly correlated to serum C-peptide (A) and insulin (B) area under the curve in 37 people with normal renal function. Regression line Spearman's $r_s$ correlations are shown. *p<0.0001.

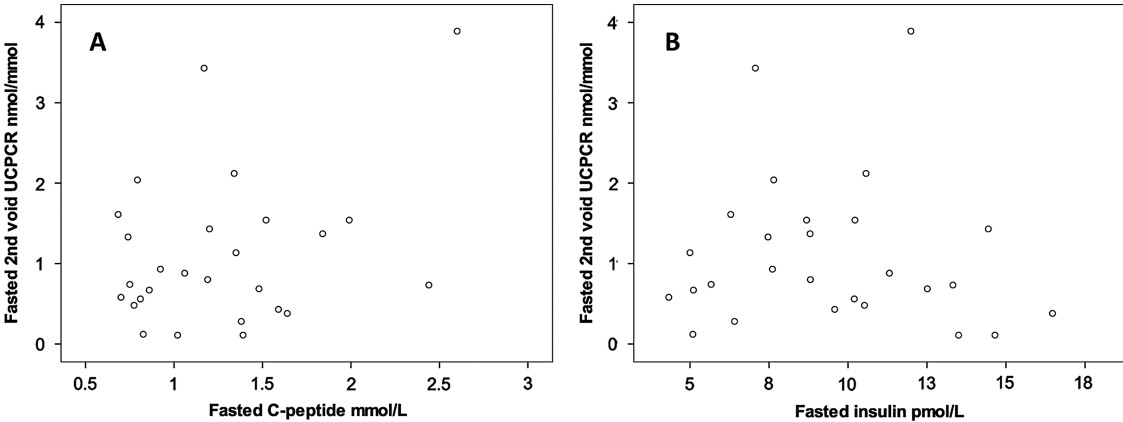

**Figure 3** Scatter plots showing no association in patients with chronic kidney disease between (A) fasting C-peptide and second-void urine C-peptide creatinine ratio (UCPCR; $r_s$ 0.17, p=0.4) and (B) fasting insulin and second-void UCPCR ($r_s$ −0.17, p=0.4).

proteinuria. Further work will be needed to fully understand the clearance of C-peptide in people with CKD. These data suggest that UCPCR should not be used in people without diabetes who have CKD. In our previously published study on patients with T2D, mild CKD (in 23 participants) did not alter the association between UCPCR and serum C-peptide.[15] It is possible that the presence of diabetes, more severe CKD (median eGFR 33 (27–46) vs 51 (44–58) in Bowman's study) or relatively small numbers in both studies may explain the difference between these two sets of results. Our results suggest that further work may be needed to assess the utility of UCPCR in participants with diabetes and renal impairment.

This study is important because of the simplicity and practicality of a UCPCR test rather than an ability to more accurately describe insulin physiology in individual participants. The current measures of insulin secretion and sensitivity rely on serum assays of C-peptide and insulin which require an access to centrifugation and freezing within 24 h. This limits studies to centres with these facilities and staff to use them. UCPCR could be particularly useful in the developing world where the diagnosis of diabetes is rising fastest and reduced facility and staffing costs associated with a posted urine sample may make large studies easier to do. Given the results with CKD and the effect of elevated glucose levels on HOMA-calculated insulin resistance, UCPCR may be most useful in young-aged or middle-aged populations where the background prevalence of CKD and diabetes is low.

In conclusion, UCPCR0 and UCPCR120 correlate with serum levels of insulin and C-peptide, and also with HOMA2-calculated insulin resistance in patients without diabetes. The practical aspects of performing UCPCR testing make it a potentially useful method for the assessment of insulin production and resistance in large epidemiology studies. Patients with CKD should be excluded from these studies.

**Contributors** RAO and AR wrote the manuscript, designed the study and performed the study. BMS was the primary statistician involved in data analysis. CB and REJB contributed in writing the manuscript, and CB recruited patients from her clinics. BAK helped design and get ethical approval for the study; she also recruited and tested patients for the study and was involved in writing the manuscript. TJMD performed all biochemical analysis of laboratory samples and contributed to writing the manuscript. ATH is the senior author and has seen multiple drafts of the paper.

**Funding** The material costs for the project were provided by a Royal Devon and Exeter NHS Foundation Trust Small Projects Grant. This study was supported by PenCLAHRC and the NIHR Exeter Clinical Research Facility.

**Competing interests** The views expressed are those of the authors and not necessarily those of the NHS, the NIHR or the Department of Health. ATH, BAK and BMS are core members of the NIHR Exeter Clinical Research Facility. ATH is supported by a Wellcome Trust Senior Investigator award(grant number 067463/Z/2/Z). RAO is a Clinical Training Fellow funded by Diabetes UK (grant number 11/0004171). TJM is an NIHR CSO Clinical Scientist Fellow. AGJ is an NIHR Doctoral Research Fellow (grant number DRF-2010-03-72). REJB was a DUK Clinical Training Fellow (grant number BDA09/0003825) when she contributed to this work.

**Ethics approval** All studies were performed with approval from the South West 2 Research Ethics Committee.

**Provenance and peer review** Not commissioned; externally peer reviewed.

**Data sharing statement** Additional data are available from Richard Oram (including individual anonymous results on UCPCR, serum blood measurements, HOMA2IR calculations and baseline characteristics of cohort. For extrainformation please write to:-r.oram@exeter.ac.uk

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
