## [Reviewer comments · BMJ Open]

Some articles will have been accepted based in part or entirely on reviews undertaken for other BMJ Group journals. These will be reproduced where possible.

ARTICLE DETAILS

TITLE (PROVISIONAL)	Urine C-Peptide Creatinine Ratio can be used to assess insulin resistance and insulin production in people without diabetes.
AUTHORS	Oram, Richard; Rawlingson, Andrew; Shields, Beverley; Bingham, Coralie; Besser, Rachel; McDonald, Timothy; Knight, Bridget; Hattersley, Andrew

VERSION 1 - REVIEW

REVIEWER	Professor Eric Kilpatrick, Hull Royal Infirmary/Hull York Medical School
	No competing interests
REVIEW RETURNED	02-Jun-2013

GENERAL COMMENTS	This study describes how a fasting, second void urine c-peptide:creatinine ratio (UCPCR) compares with other, more traditional, measures of insulin resistance in non-diabetic subjects with and without CKD. They found that UCPCR related to the other measures in subjects with normal renal function, but not in those with CKD. General Points  1. While the logistics of separating and freezing serum insulin samples can certainly be difficult, it seems that a fasting, second voided urine may be as well. Indeed, if a serum creatinine and a fasting glucose measurement have to be performed to exclude those with CKD and glucose intolerance, is it not simpler just to measure fasting insulin at the same time? 2. The authors feel this test could be used in population screening, but would the fact that a third of normal elderly subjects can have an eGFR<60 not limit its use somewhat? Likewise, having to exclude diabetes could be a major limitation too. 2. Why was the relationship with the first voided urine not presented? At least this would be easier for the patient to provide. 3. The relationships between fasting UCPCR and other tests only accounts for under 50% of the variance (r-squared) of one another. Is that likely to be good enough? Specific points Line numbers as in pdf Page Line 3 35 It might be helpful for the general audience to explain why a urine insulin:creatinine ratio is not feasible 5 10 How long after the first void was the second one taken? What would happen if the patient did not wish to 'go' a second time? 9 27 'not known' should read 'known not' 9 50 As stated above, if a fasting glucose measurement is required to exclude glucose intolerance, why not just measure insulin at the same time?
---

	10 15 As stated above, why did the first void sample not have UCPCR measured here? 11 41 R Besser is not included as an author in the first page of submission
--	--

REVIEWER	Sattar, Naveed University of Glasgow, BHF centre
REVIEW RETURNED	19-Jun-2013

THE STUDY	the association of UCPCR to insulin and HOMA-IR are of interest but still R2 values would be only around 50% - This group have recently published evidence in BMJ open to show insulin measurements are unaffected at room temperature for 24 hours - this paper should be mentioned for balance EDTA improves stability of whole blood C-peptide and insulin to over 24 hours at room temperature. McDonald TJ, Perry MH, Peake RW, Pullan NJ, O'Connor J, Shields BM, Knight BA, Hattersley AT. PLoS One. 2012;7(7):e42084. doi: 10.1371/journal.pone.0042084. Epub 2012 Jul 30.
RESULTS & CONCLUSIONS	I would have liked to see the relative comparison of UCPCR to HOM-IR or insulin versus simpler lifestyle measures so age, gender, BMI, waist, etc - and lipids if they have them In other words, does UCPCR associate with fasting insulin independent of simple measures of insulin resistance which are easily obtained since if it does, then this helps but if it does not, then their conclusion would potentially require a bigger study OR else, they down play conclusion and say that evaluation of their data suggests more studies should look at this measure as a potential surrogate of IR in larger studies and give caveats and limitations more openly
REPORTING & ETHICS	not sure CONSORT needed as methodology related?
GENERAL COMMENTS	this is nice study from excellent group but I think the data cannot currently be used to make such a strong statement as "A novel method for the assessment of insulin resistance and insulin production in people without diabetes". I would temper statements and call this a pilot study and suggest how work could be taken forwards

VERSION 1 – AUTHOR RESPONSE

Reviewer: Professor Eric Kilpatrick, Hull Royal Infirmary/Hull York Medical School

This study describes how a fasting, second void urine c-peptide:creatinine ratio (UCPCR) compares with other, more traditional, measures of insulin resistance in non-diabetic subjects with and without CKD. They found that UCPCR related to the other measures in subjects with normal renal function, but not in those with CKD.

General Points

1. While the logistics of separating and freezing serum insulin samples can certainly be difficult, it seems that a fasting, second voided urine may be as well. Indeed, if a serum creatinine and a fasting glucose measurement have to be performed to exclude those with CKD and glucose intolerance, is it not simpler just to measure fasting insulin at the same time?

ANSWER:-In our previous studies we have found that a fasted second-void sample can be done by people of all ages without difficulty. A fasting glucose and creatinine to rule out renal impairment and diabetes would negate some of the advantages of UCPCR, we have added a phrase in the conclusion suggesting that UCPCR may be most useful in populations with a low background prevalence of CKD and diabetes. We agree that fasting insulin may be simple if the storage and processing of samples is possible but still requires a visit to a hospital facility and staff to take the blood test.

2. The authors feel this test could be used in population screening, but would the fact that a third of normal elderly subjects can have an eGFR<60 not limit its use somewhat? Likewise, having to exclude diabetes could be a major limitation too.

ANSWER:- We agree that this test would be most suitable for young or middle aged populations where the background prevalence of CKD is low and have added this to the discussion. Demonstrating that UCPCR is not seem useful in those with CKD (without diabetes) is an important limitation for people thinking of using UCPCR as a method.

3. Why was the relationship with the first voided urine not presented? At least this would be easier for the patient to provide.

ANSWER:- We did not collect first void urines as first void samples were more variable in a previous study, we have included this in the methods now (McDonald et al Clin Chem 2009). We comment in the discussion that if there are datasets/studies out there with fasted first void urine available, this may be an interesting question to address.

4. The relationships between fasting UCPCR and other tests only accounts for under 50% of the variance (r-squared) of one another. Is that likely to be good enough?

ANSWER:- In epidemiology studies comparing large populations, e.g. between different racial groups or to detect a change in insulin resistance after an intervention, a measure that accounts for 48% of variation will still be useful. We are not advocating this test for individual use where a variance at this level may not be informative enough.

Specific points Line numbers as in pdf

Page Line

3 35 It might be helpful for the general audience to explain why a urine insulin:creatinine ratio is not feasible

ANSWER:- Paragraph changed, "unlike insulin" added

5 10 How long after the first void was the second one taken? What would happen if the patient did not wish to 'go' a second time?

ANSWER:- The length of time between the first and second void was not recorded. It was only documented that the patient had voided their bladder at home prior to starting the study. This has been amended in the methods.

9 27 'not known' should read 'known not'

ANSWER:- changed

9 50 As stated above, if a fasting glucose measurement is required to exclude glucose intolerance, why not just measure insulin at the same time?

ANSWER:- Answered above

10 15 As stated above, why did the first void sample not have UCPCR measured here?

ANSWER:- Patients voided their bladder at home but did not keep the sample – methods changed as above.

11 41 R Besser is not included as an author in the first page of submission

ANSWER:- changed

Reviewer: Naveed Sattar
University of Glasgow, BHF centre

the association of UCPCR to insulin and HOMA-IR are of interest but still R2 values would be only around 50% -

ANSWER:- We agree that UCPCR only explains approximately 50% of the variation of HOMA-IR, and this may affect whether it could be used at an individual level. But other simple measures such as BMI account for less variation and can still be useful at a population level.

This group have recently published evidence in BMJ open to show insulin measurements are unaffected at room temperature for 24 hours - this paper should be mentioned for balance
EDTA improves stability of whole blood C-peptide and insulin to over 24 hours at room temperature.
McDonald TJ, Perry MH, Peake RW, Pullan NJ, O'Connor J, Shields BM, Knight BA, Hattersley AT. PLoS One. 2012;7(7):e42084. doi: 10.1371/journal.pone.0042084. Epub 2012 Jul 30.

ANSWER:- We have included this point and the reference in the introduction

I would have liked to see the relative comparison of UCPCR to HOM-IR or insulin versus simpler lifestyle measures

so age, gender, BMI, waist, etc - and lipids if they have them

ANSWER:- Thank you for this suggestion. Both age and BMI correlated with HOMA2IR (spearman's correlation 0.50 and 0.52 respectively. We did not have lipids and waist circumference in this group. This is less than the correlation of UCPCR with HOMA-IR (r^2 of 0.25 and 0.27 compared to an $r^2=0.48$ for UCPCR0). We have added this to the results and the discussion.

In other words, does UCPCR associate with fasting insulin independent of simple measures of insulin resistance which are easily obtained since if it does, then this helps but if it does not, then their conclusion would potentially require a bigger study

ANSWER:- Hopefully this is answered above.

OR else, they down play conclusion and say that evaluation of their data suggests more studies should look at this measure as a potential surrogate of IR in larger studies and give caveats and limitations more openly

ANSWER:- See above

this is nice study from excellent group but I think the data cannot currently be used to make such a strong statement as "A novel method for the assessment of insulin resistance and insulin production in people without diabetes". I would temper statements and call this a pilot study and suggest how work could be taken forwards.

ANSWER:- We have toned down the title

VERSION 2 – REVIEW

REVIEWER	Professor Eric Kilpatrick, Hull Royal Infirmary/Hull York Medical School, UK
REVIEW RETURNED	12-Aug-2013

GENERAL COMMENTS	I suspect the authors have answered my points as well as is possible given the data, so from that perspective it is an acceptable paper, but it is debatable how practical and reliable this test would be, even in a population study.
---